# High-Resolution Drone-Acquired RGB Imagery to Estimate Spatial Grape Quality Variability

**Marta García-Fernández** * , **Enoc Sanz-Ablanedo** and **José Ramón Rodríguez-Pérez**

Grupo de Investigación en Geomática e Ingeniería Cartográfica (GEOINCA), Universidad de León, Avenida de Astorga sn, 24401 Ponferrada, León, Spain; esana@unileon.es (E.S.-A.); jr.rodriguez@unileon.es (J.R.R.-P.)
* Correspondence: mgarcf@unileon.es

**Abstract:** Remotesensing techniques can help reduce time and resources spent collecting samples of crops and analyzing quality variables. The main objective of this work was to demonstrate that it is possible to obtain information on the distribution of must quality variables from conventional photographs. Georeferenced berry samples were collected and analyzed in the laboratory, and RGB images were taken using a low-cost drone from which an orthoimage was made. Transformation equations were calculated to obtain absolute reflectances for the different bands and to calculate 10 vegetation indices plus two new proposed indices. Correlations for the 12 indices with values for 15 must quality variables were calculated in terms of Pearson's correlation coefficients. Significant correlations were obtained for 100-berries weight (0.77), malic acid ($-0.67$), alpha amino nitrogen ($-0.59$), phenolic maturation index (0.69), and the total polyphenol index (0.62), with 100-berries weight and the total polyphenol index obtaining the best results in the proposed RGB-based vegetation index 2 and RGB-based vegetation index 3. Our findings indicate that must variables important for the production of quality wines can be related to the RGB bands in conventional digital images, potentially improving and aiding management and increasing productivity.

**Keywords:** remotesensing; drone; RGB imagery; spectral index; vineyard zoning; must quality variable

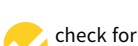



## 1. Introduction

Viticulture is an important economic sector in Spain, which, with Italy, France and China, tops international lists in terms of surface area and production (OIV, 2014) [1]. On the banks of the Duero River, in Spain, the Ribera del Duero Protected Denomination of Origin (DO) extends over 20,000 ha and produces an average of 90,000 tons per year of Tempranillo (95% of harvest volume), Cabernet, Sauvignon, Merlot, and Malbec grape varieties. The vineyards belonging to the Ribera del Duero DO, at an average altitude of 850 m above sea level, are characterized by soils of different tertiary sediments, mainly formed by layers of clay, limestone, and sand. The clay layer, usually located in lower areas closer to the River Douro, is warmer and more permeable, and, therefore, richer in nutrients. The temperature of the limestone layer, located at higher altitudes, is lower, due to high reflectance, less permeability, and less moisture retention. The Ribera del Duero climatology, according to the Papadakis agroclimatic classification, is hot and dry temperate continental Mediterranean [2], and is featured by moderate-low rainfall (400–600 mm), dry summers (summer aridity), harsh winters (January averages below 3 °C and frost-free periods of less than 140 days), and marked thermal oscillation [3]. Due to the characteristics of the soil, some vines suffer from nutritional iron deficiency (iron chlorosis), occurring mainly in limestone or high-pH soils where the availability of this microelement is very low [4]. Relatively high berry temperatures (30 °C) are essential for the production of quality wines [5] and depend not only on climatic and topographic factors (such as altitude and orientation), but also on canopy size, vine density, and crop vigor [6,7]. Properties that stand out for their importance in producing higher quality wines are contents of flavonoids (tannins (TAN), anthocyanins (ANT), and flavanols), total acidity (ToA), total

soluble solids (TSS), pH, alpha amino nitrogen (AAN), easily assimilated nitrogen (EAN), and gluconic acid (GA). Grapes that are relatively less exposed to sunlight have different quality values than more exposed grapes, whose increased temperature and different color result in different phenolic contents [6]. The highest concentrations of flavonoids, which significantly influence wine aroma and flavor, occur at higher temperatures [6]. ToA, TSS, and pH are balanced in the maturity stage when canopy temperatures are between 18 °C and 30 °C [5]. AAN, EAN, and GA influence wine aroma. When physiological maturity is achieved, sugar concentration is high, while maintaining an adequate canopy temperature is necessary for avoiding nitrogen loss.

A main objective of precision viticulture is to determine spatial variations in the quality parameters of must and manage a crop at the parcel level [8], as increasing production values contribute to more efficient and accurate management of inputs [9–12]. Must quality is usually estimated through repeated sampling during the berry maturation process [13], which, however, is an impractical approach for large areas [14,15]. Several studies highlighted the possibility of obtaining crop information using remotesensing tools, especially aircraft, satellites, or drones (unmanned aerial vehicles, UAVs). Nevertheless, for grapevine, plant discontinuity and zones of shade greatly affect the calculation of spectral indices [16–18], making the use of remote sensing techniques challenging [19]. Taking these limitations into account, viticulture researchers minimized the influence of unwanted areas using digital image processing techniques. In this way, they obtain results suitable for monitoring, analyzing, and mapping variations in vegetation structure and biophysical parameters [14,19–21] with a goal of predicting quality variables [6,22–27] and detecting pests and diseases [28–30]. In several studies, the potential of satellite images such as Landsat 8 (ground sample distance (GSD) = 30 m) [19], WorldView-2, and Pleiades (GSD = 0.5 m and 2 m) [20,30], Sentinel-2 (GSD = 2 m and 10 m) [30–33], and remote multispectral high-resolution images (GSD = 0.2–0.6 m) [6,22–25] were evaluated using digital image processing tools that showed good correlations as a function of the heterogeneity of vineyards, data, and acquisition times [19]. These also showed strong correlations in estimations of the Brix, phenolic content, pH, ANT, berry weight, and acidity variables, using an image classification threshold to eliminate spectral information on bare soil and the areas between the rows [6,22–27].

Recent advances in drone and sensor technologies brought improvements to the viticulture field in the form of high-resolution images of centimetric precision, time and cost reductions for information acquisition, greater ease and flexibility in sensor exchanges, and in working time planning [14,34–36]. Using tools based on digital image processing techniques such as structure-from-motion matching and segmentation, several authors assessed the possibility of accurately obtaining information on spatial variability in grape quality parameters, vigor zones, differentiated irrigation areas, and vine-related diseases. The use of high-resolution images improved the creation of filtered maps depicting correlations with vigor [14,21], allowed missing plants to be counted [14], provided good correlations with soluble solids and pH [25,26], and provided good prediction models for pest detection [28–30] and the classification of different irrigation areas [37]. Despite the variety of applications based on the use of UAV-captured images, little research is done in the viticulture field to apply RGB-based vegetation indices to the prediction of must quality.

For our research we used vegetation indices created from RGB bands captured by conventional cameras installed in small UAVs, as shown to be useful in other precision viticulture studies [14,28–30,37]. The main objective was to identify which RGB-based vegetation indices could be used in a meaningful way to estimate variables related to must quality parameters, as determined by the usual analytical techniques in enological laboratories. The analyzed indices include ten existing vegetation indices and two developed by the authors. RGB images captured by a low-cost UAV were processed to extract information on vegetation canopy reflectance and spectral indices, and to determine the Pearson correlation coefficient with values for must quality variables.

## 2. Materials and Methods

### 2.1. Methodology

The steps of the methodology, depicted in Figure 1, were as follows—(1) collection of georeferenced samples of grape berries, (2) RGB capture by UAV, (3) laboratory analysis of sampled must, (4) digital image processing to obtain an orthoimage of the study area, (5) transformation of digital numbers to reflectances, (6) calculation of RGB-based vegetation indices, (7) analysis of correlations between must variables and vegetation indices, and (8) spatial mapping of variability in quality variables.

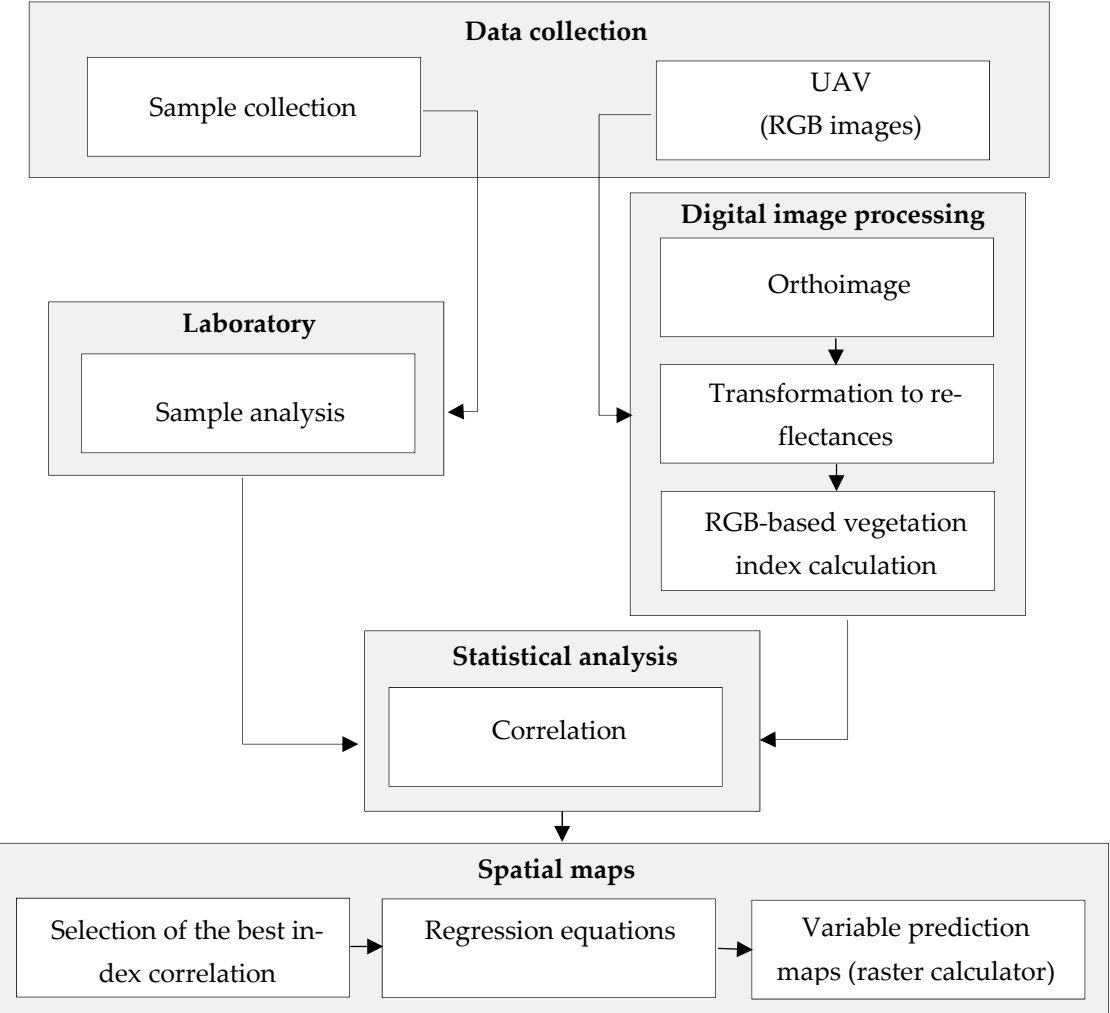

**Figure 1.** Flowchart illustrating the study methodology.

### 2.2. Study Area

Fieldwork was carried out in the Finca Martinete commercial Ribera del Duero DO vineyard (cv. Tempranillo; Olivares del Duero, Valladolid, Spain) planted in 1997 (Figure 2), with centerpoint coordinates 383800 4610100 CRS UTM30N/ETRS89.

An area of 5.83 ha was cultivated in trellis lines, oriented northwest to southeast, with a separation between rows of 3 m. The parcel, with soil of a clayey texture, was at an altitude of 725–750 m and had a slope of 0–15%. Due to the characteristics of the soil, some of the vines grown on this plot experienced nutritional iron deficiency (iron chlorosis).

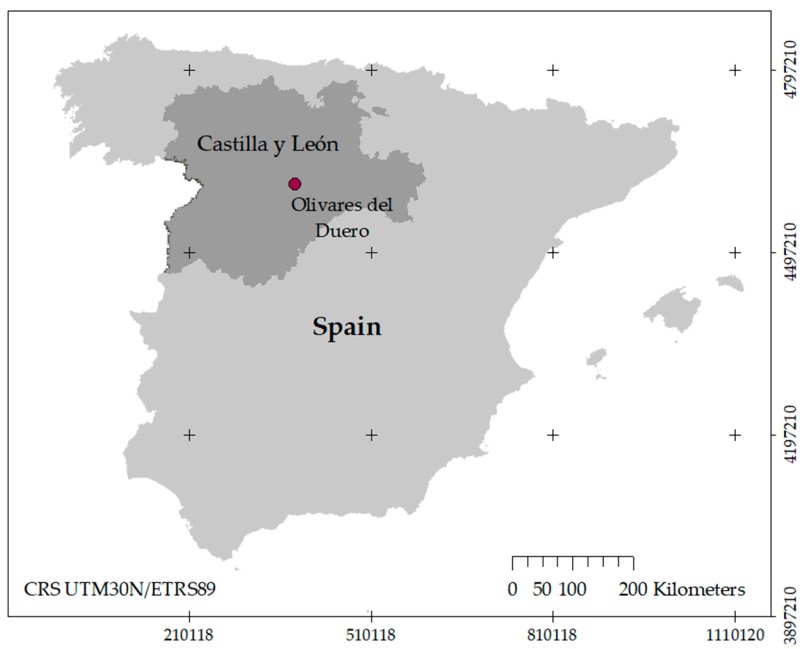

**Figure 2.** Olivares del Duero work area location, with center point coordinates 383800 4610100 CRS UTM30N/ETRS89.

### 2.3. Field Sampling and Analytical Testing

The experimental area, located in the southeastern part of the vineyard, was regularly sampled (Figure 3a). Excluding the first three rows of the crop, four longitudinal strips were defined, consisting of two consecutive vine lines, with each strip separated by two rows of cultivation. Three transversal repetitions were defined in each strip to form 12 blocks (Figure 3b) from which the berries were sampled.

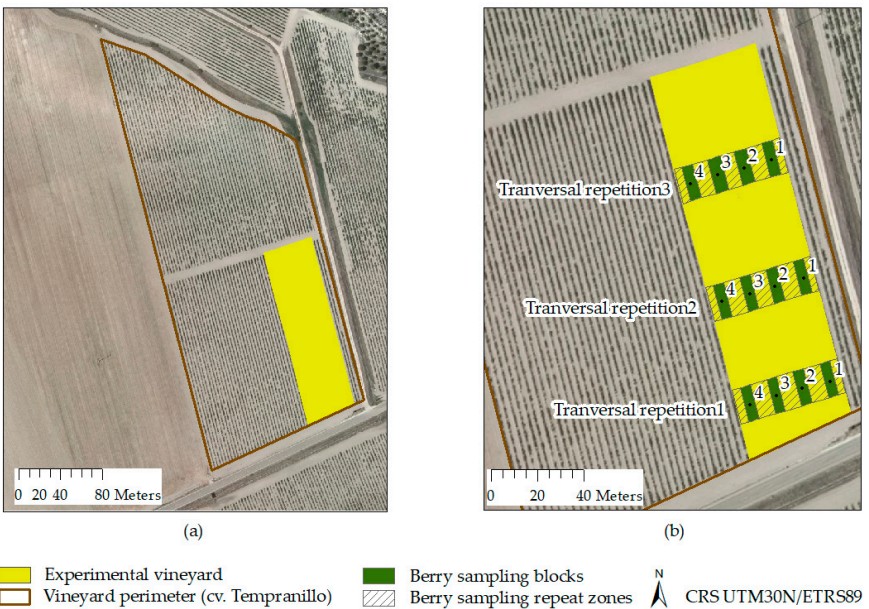

**Figure 3.** Finca Martinete vineyard (cv. Tempranillo; Olivares del Duero, Valladolid, Spain). (**a**) Vineyard with experimental area. (**b**) Sampling blocks and repeat zones.

From each of the 12 sampling blocks, 34 of the most representative berries were sampled, for a total of 408 berries per block, then stored in labeled bags for refrigerated transport to the laboratory, where they were analyzed by winery technicians for the 15 quality vari-

ables described in Table 1. The samples were georeferenced using a Global Navigation Satellite System (GNSS) receiver (Leica Viva GNSS GS08 Plus, LEICA Geosystems AG).

**Table 1.** Fifteen laboratory-analyzed must quality variables.

| Acronym | Variable | Units | Instrumentation |
|---|---|---|---|
| BW | 100-berry weights | $(kg \cdot 10^{-3})$ | Precision scale |
| MA | Malic acid | (g/L) | OENO FOSS |
| TaA | Tartaric acid | (g/L) | OENO FOSS |
| AAN | Alpha amino nitrogen | (mg/L) | OENO FOSS |
| EAN | Easily assimilated nitrogen | (mg/L) | OENO FOSS |
| GA | Gluconic acid | (g/L) | OENO FOSS |
| TSS | Total soluble solids | (°Bx) | OENO FOSS |
| ToA | Total acid | (g/L) | OENO FOSS |
| pH | pH | (pH) | OENO FOSS |
| PSC | Probable stable color | | CROMOENO |
| PRI | Phenolic ripeness index | | CROMOENO |
| TPI | Total phenolic index | | CROMOENO |
| PCAF | Probable color by end of alcoholic fermentation | | CROMOENO |
| ANT | Anthocyanins | | CROMOENO |
| TAN | Tannins | | CROMOENO |

*2.4. Drone Image Acquisition*

Drone images were captured in early September 2017, at 12 p.m. local time (UTC+2 in daylight saving time) in optimal weather conditions, using a DJI Phantom 3 Professional UAV (SZ DJI Technology Co, Ltd., Shenzhen, China) with a DJI FC300X preinstalled sensor (Figure 4). Flight programming was done using a Pix4D Capture (Pix4D S.A) Android application installed on a smartphone. The images were captured in six longitudinal, passes at a height of 26 m for a GSD of 1 cm, with longitudinal and transverse overlaps of 80% and 60%, respectively. In total, 173 images were taken with preset exposure parameters (aperture, exposure time, ISO sensitivity, and white balance). To georeference the photogrammetric model, eight control points were materialized in the field. The coordinates were taken in the RTK mode with a centimeter-precision GNSS receiver and with corrections from the RTCM 3.0 network (VRS3).

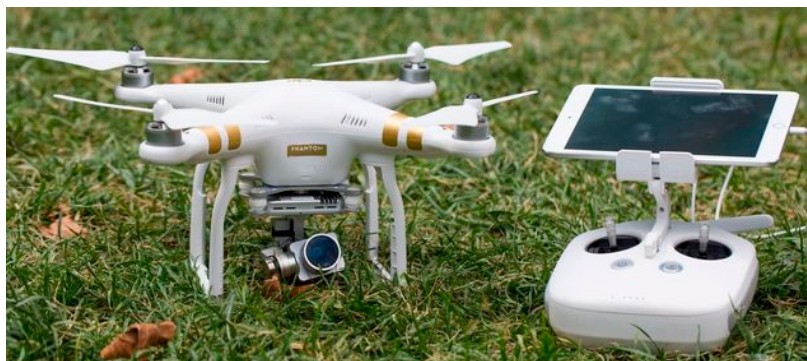

**Figure 4.** DJI Phantom 3 Professional UAV.

*2.5. Orthoimage*

Block orientation was calculated using the Agisoft Photoscan Software V1.3.1. (Agisoft LLC., St. Petersburg, Russia) structure-from-motion technique. Prior to aligning the images, capture quality was analyzed to eliminate blurry images. Using the automatic image quality estimation function from the same software, the sharpness index was calculated and images with values indicating poor quality (<0.75) were eliminated from the analysis.

The images were oriented using the medium precision option, achieving, in a first step, a low-resolution model that coincided with the alignment of all images. In a second step, image match alignment was repeated at a higher resolution, considering only the images paired with the overlapping areas. For the alignment calculations, a search limit of 40,000 points was set to use the 4000 highest quality link points.

For georeferencing purposes, all control points were manually measured using a crosshair pointer. An autocalibration mathematical model was used to fit the entire image block. This process resulted in a dense point cloud, focal distance, displacement of the main point, four radial distortion coefficients, four tangential distortion coefficients, two affinity and non-orthogonality (bias) coefficients, and image widths and heights. A digital elevation model (DEM) was created from the dense point cloud and the images were rectified using an indirect transformation based on the DEM data. The digital level (DL) assigned to each position was the average value of all pixels in the original images.

### 2.6. Orthoimage Transformation to Reflectances

Calculating the absolute reflectance values from the DL captured by a sensor is an important aspect of remote data detection, since it makes it possible to obtain quantitative values for spectral reflectance and to obtain results that are comparable with other works.

The transformation of the DL of each band of the orthoimage to reflectance was performed with Equations (1)–(3) for the R, G and B bands, respectively, where $\lambda$ (R), $\lambda$ (G), and $\lambda$ (B) were the outcome of transforming the RGB bands of the orthoimage to reflectances, and DL (R), DL (G), and DL (B) were the RGB bands of the orthoimage in DL.

$$\lambda(R) = 0.013089e^{0.015672 \times DL(R)} \tag{1}$$

$$\lambda(G) = 0.018661e^{0.014890 \times DL(G)} \tag{2}$$

$$\lambda(B) = 0.019370e^{0.014933 \times DL(B)} \tag{3}$$

The equations were obtained by means of an analysis of correlations between reflectances and their corresponding DLs. A total of 41 gray level panels were placed during flight and then captured in various images, on which the reflectance measurements were made using a portable ASD FieldSpec4 spectroradiometer (Analytical Spectral Devices, Inc., Boulder, CO, USA). On these images, the DL of each panel was measured in the RGB bands and adjusted by means of Equations (1)–(3), at wavelengths corresponding to maximum RGB sensitivity. In small-format multispectral cameras, such as those used in UAV flights for precision agriculture, spectral response information is usually limited. This information is important, however, in the process of converting DLs to reflectivity values in the different channels of the camera [38], in the empirical line method of radiometric calibration. To determine the sensitivity peaks of the sensor in each band, the correlation analysis was performed between the reflectances measured on targets of various shades and the corresponding DLs extracted from photographs. The results showed maximum correlation in wavelengths 660 nm for R, 545 nm for G, and 470 nm for B. The sensor characterization process is described elsewhere [39].

To transform each band of the orthoimage to reflectances, the ArcGIS v10.4.1 software (ESRI Inc., Redlands, CA, USA) map-algebra option was used with Equations (1)–(3) for bands R, G, and B, respectively.

### 2.7. Reflectance Values

Reflectance values for the vineyard canopy were extracted in order to determine the correlations between the berry quality variables and reflectance of the vegetation located in the sampling areas. To select the vegetation canopy, supervised classification was performed to isolate the pixels with the RGB values from the vine lines. To measure the localized reflectance values in the sampling areas, a vector format layer of circumference 192 (16 × 12) zones and radius 0.10 m was created using the ArcGIS v10.4.1 software. Using

the zonal-statistics-as-table algorithm of the same software and the previously created circumference layer, the average RGB reflectance values for the areas were calculated in table format.

### 2.8. Vegetation Indices

The use of vegetation indices instead of reflectance is a common practice in remote-sensing studies, as they minimize the influence of distorting factors, such as the ground, solar irradiance, the elevation angle of the sun, and the atmosphere itself [40]. In this study, 12 vegetation indices were selected and explored, based on the RGB color space and their correlations with the 15 must quality variables; described in Table 1. Of the analyzed indices, ten were selected from the literature (listed in the first ten rows of Table 2) and two indices were developed by the authors (the last two rows of Table 2). The new RGB-based vegetation index 2 (RGBVI2) and RGB-based vegetation index 3 (RGBVI3) were developed on the basis of different combinations of the RGB bands. The criterion for the new vegetation indices was to try to discover the relationships between the three bands with the best results. Thus, all possible combinations were tested and the indices with the most prominent correlations were selected, some already published and others as proposed for this study. Note that the blue band was included since it contributes to enhancing the differences between leaf proportions reflecting different vegetative states [41]. The indices used were as follows—the simple red–green ratio (GR) [42]; the red–green vegetation index (GRVI) [43]; the new red–green–blue vegetation index (RGBVI) and the modified GRVI (MGRVI) [21] reflecting vine vigor and biomass; the visible atmospheric resistance index (VARI) [44,45]; the simple blue–green ratio index (BGI$_2$) [18] reflecting chlorophyll content and vigor; the vegetative index (VEG) [46]; the green leaf index (GLI) [47] and the excess green index (ExG) [48] that automatically detected areas related to bare soil, weeds, residues [37], and water stress conditions [29]; and the normalized green–blue difference index (NGBDI) that reflected visualization of changes in growth states [49].

**Table 2.** Twelve vegetation indices used in this research.

| Acronym | Indices | Definition | Author and Year |
|---------|---------|------------|-----------------|
| GR | Simple red–green ratio | $\frac{R}{G}$ | [42] |
| GRVI | Green–red vegetation index | $\frac{G-R}{G+R}$ | [43] |
| RGBVI | RGB-based vegetation index | $\frac{G^2-(B\times R)}{G^2+(B\times R)}$ | [21] |
| MGRVI | Modified green–red vegetation index | $\frac{G^2-R^2}{G^2+R^2}$ | [21] |
| VARI | Visible atmospherically resistant index | $\frac{G-R}{G+R-B}$ | [44] |
| BGI$_2$ | Simple blue–green ratio | $\frac{B}{G}$ | [18] |
| VEG | Vegetativen | $\frac{G}{R^a \times B^{(1-a)}}$ ; a = 0.667 | [46] |
| GLI | Green leaf | $\frac{2G-R-B}{2G+R+B}$ | [47] |
| ExG | Excess green index | $2G-R-B$ | [48] |
| NGBDI | Normalized green-blue difference index | $\frac{G-B}{G+B}$ | [49] |
| RGBVI2 | RGB-based vegetation index 2 | $\frac{G-R}{B}$ | Proposed |
| RGBVI3 | RGB-based vegetation index 3 | $\frac{G+B}{R}$ | Proposed |

### 2.9. Statistical Analysis

Statistical methods were used to calculate the correlations between the grape quality variables and the vegetation indices, to obtain regression equations for the grape quality variables and the vegetation indices for statistically significant correlations, and to spatially map the quality variable distributions on the vineyard surface.

SPSS v.17.0 software (SPSS Inc., Chicago, IL, USA) was used to perform the exploratory analysis of the vegetation indices. Minimums, maximums, means, standard deviations, and variances of vegetation indices were calculated. To designate outliers, a threshold was established at three times the standard deviation value and any values that exceeded the calculated limits were removed.

The 12 vegetation indices (Table 2) were calculated with the average RGB reflectance values for the sampling areas (obtained as described in Section 2.7). Using those vegetation indices and the data obtained from the quality variables of the resampled berries (Table 3, commented below), Pearson's correlation coefficients were calculated (Table 4). Only the most significant correlations were selected for the purpose of calculating the regression lines that related the vegetation indices with the quality variables. Using the regression equations and the ArcGIS v10.4.1 software raster calculator tool, spatial distribution maps were created for the variables that showed significant relationships with the RGB vegetation indices.

**Table 3.** Laboratory results for 15 must quality variables.

| Z | R | BW | MA | TaA | AAN | EAN | GA | SST | ToA | pH | PSC | PRI | TPI | PCAF | ANT | TAN |
|---|---|----|----|-----|-----|-----|----|-----|-----|----|-----|-----|-----|------|-----|-----|
| 1 | 1 | 112.0 | 2.3 | 9.3 | 342.7 | 252.1 | 0.2 | 20.7 | 3.93 | 3.80 | 7.30 | 1.63 | 43.07 | 10.12 | 1712.0 | 1167.8 |
| 1 | 2 | 107.0 | 2.3 | 9.5 | 332.8 | 239.3 | 0.2 | 20.9 | 3.92 | 3.80 | 7.20 | 1.66 | 43.12 | 10.57 | 1753.5 | 1221.0 |
| 1 | 3 | 104.0 | 2.4 | 9.8 | 345.7 | 254.2 | 0.5 | 21.0 | 3.81 | 3.95 | 7.30 | 1.62 | 42.75 | 10.95 | 1780.4 | 1152.7 |
| 2 | 1 | 120.0 | 2.0 | 9.1 | 237.2 | 214.2 | 0.3 | 20.6 | 3.94 | 3.70 | 7.01 | 1.72 | 42.12 | 10.55 | 1612.4 | 1122.3 |
| 2 | 2 | 115.0 | 2.2 | 9.0 | 241.2 | 205.7 | 0.2 | 20.9 | 3.93 | 3.80 | 6.97 | 1.70 | 42.20 | 10.63 | 1553.6 | 1152.7 |
| 2 | 3 | 112.0 | 2.0 | 8.9 | 235.3 | 203.6 | 0.2 | 20.9 | 3.94 | 3.80 | 6.94 | 1.75 | 42.16 | 10.58 | 1624.4 | 1138.3 |
| 3 | 1 | 121.0 | 2.2 | 9.0 | 263.4 | 219.8 | 0.2 | 20.6 | 3.87 | 3.70 | 7.40 | 1.68 | 42.12 | 11.08 | 1733.2 | 1118.9 |
| 3 | 2 | 113.0 | 2.1 | 8.9 | 257.1 | 227.7 | 0.3 | 20.7 | 3.97 | 3.70 | 7.30 | 1.72 | 44.00 | 11.03 | 1712.0 | 1152.3 |
| 3 | 3 | 112.0 | 2.1 | 9.1 | 261.2 | 221.9 | 0.8 | 20.8 | 3.99 | 3.80 | 7.40 | 1.70 | 43.82 | 11.06 | 1730.0 | 1178.8 |
| 4 | 1 | 121.0 | 1.9 | 9.0 | 219.2 | 205.6 | 0.5 | 20.7 | 3.94 | 3.75 | 7.50 | 1.81 | 45.70 | 11.06 | 1722.0 | 1218.5 |
| 4 | 2 | 116.0 | 1.8 | 8.9 | 220.3 | 212.5 | 0.6 | 21.2 | 4.02 | 3.80 | 7.35 | 1.73 | 45.32 | 11.08 | 1635.2 | 1135.8 |
| 4 | 3 | 130.0 | 1.9 | 9.0 | 215.6 | 206.5 | 0.6 | 21.3 | 4.01 | 3.75 | 7.42 | 1.82 | 45.26 | 11.09 | 1672.4 | 1213.9 |

Z: sampling zone; R: repeat sampling zone; BW: 100-berry weight (kg·$10^{-3}$); MA: malic acid (g/L); TaA: tartaric acid (g/L); AAN: alpha amino nitrogen (mg/L); EAN: easily assimilated nitrogen (mg/L); GA: gluconic acid (g/L); TSS: total soluble solids (°Bx); ToA: total acid (g/L); pH (pH); PSC: probable stable color; PRI: phenolic ripeness index; TPI: total phenolic index; PCAF: probable color by end of alcoholic fermentation (CROMOENO data); ANT: anthocyanin content (CROMOENO data); and TAN: tannin content (CROMOENO data).

**Table 4.** Pearson linear correlations between 15 must quality variables and 12 vegetation indices.

| | GR | GRVI | RGBVI | MGRVI | VARI | BGI$_2$ | VEG | GLI | ExG | NGBDI | RGBVI2 | RGBVI3 |
|------|------|------|-------|-------|------|------|------|------|------|-------|--------|--------|
| BW | −0.76 ** | 0.76 ** | 0.52 | 0.76 ** | 0.76 ** | 0.56 | 0.71 ** | 0.72 ** | 0.21 | −0.12 | 0.77 ** | 0.72 ** |
| MA | 0.66 * | −0.66 * | −0.33 | −0.66 ** | −0.67 * | −0.59 * | −0.54 | −0.55 | 0.00 | 0.25 | −0.65 * | −0.67 * |
| TaA | 0.50 | −0.50 | −0.23 | −0.50 | −0.50 | −0.45 | −0.39 | −0.41 | 0.04 | 0.21 | −0.49 | −0.51 |
| AAN | 0.58 * | −0.59 * | −0.43 | −0.59 * | −0.59 * | −0.42 | −0.56 | −0.56 | −0.09 | 0.07 | −0.58 * | −0.55 |
| EAN | 0.46 | −0.46 | −0.41 | −0.46 | −0.46 | −0.28 | −0.48 | −0.48 | −0.12 | −0.04 | −0.46 | −0.41 |
| GA | −0.05 | 0.07 | −0.05 | 0.06 | 0.07 | 0.12 | 0.02 | 0.01 | −0.09 | −0.12 | 0.05 | 0.10 |
| TSS | −0.22 | 0.23 | 0.47 | 0.23 | 0.22 | −0.09 | 0.41 | 0.39 | 0.53 | 0.36 | 0.25 | 0.11 |
| ToA | −0.33 | 0.34 | 0.24 | 0.33 | 0.33 | 0.24 | 0.33 | 0.32 | 0.19 | −0.04 | 0.34 | 0.32 |
| pH | 0.52 | −0.52 | −0.32 | −0.52 | −0.51 | −0.40 | −0.45 | −0.47 | −0.04 | 0.12 | −0.52 | −0.49 |
| PSC | −0.37 | 0.39 | −0.03 | 0.38 | 0.39 | 0.52 | 0.18 | 0.19 | −0.26 | −0.43 | 0.36 | 0.47 |
| PRI | −0.68 * | 0.69 * | 0.48 | 0.69 * | 0.69 * | 0.52 | 0.64 * | 0.65 * | 0.06 | −0.11 | 0.68 * | 0.66 * |
| TPI | −0.57 | 0.59 * | 0.24 | 0.59 * | 0.59 * | 0.57 | 0.45 | 0.45 | −0.02 | −0.29 | 0.56 | 0.62 * |
| PCAF | −0.42 | 0.43 | 0.37 | 0.43 | 0.43 | 0.26 | 0.44 | 0.44 | 0.02 | 0.03 | 0.42 | 0.39 |
| ANT | 0.14 | −0.14 | −0.28 | −0.14 | −0.13 | 0.05 | −0.24 | −0.23 | −0.37 | −0.21 | −0.15 | −0.06 |
| TAN | −0.08 | 0.09 | 0.15 | 0.09 | 0.09 | 0.01 | 0.15 | 0.13 | 0.11 | 0.10 | 0.08 | 0.07 |

Significance: ** $p < 0.01$; * $p < 0.05$. **Vegetation indices.** GR: simple red-green ratio; GRVI: green-red vegetation index; RGBVI: RGB-based vegetation index; MGRVI: modified green-red vegetation index; VARI: visible atmospherically resistant index; BGI$_2$: simple blue-green ratio; VEG: vegetativen; GLI: green leaf index; ExG: excess green index; NGBDI: normalized green-blue difference index; RGBVI2: RGB-based vegetation index 2; RGBVI3: RGB-based vegetation index 3. **Must quality variables.** BW: 100-berry weight (kg·$10^{-3}$); MA: malic acid (g/L); TaA: tartaric acid (g/L); AAN: alpha amino nitrogen (mg/L); EAN: easily assimilated nitrogen (mg/L); GA: gluconic acid (g/L); TSS: total soluble solids (°Bx); ToA: total acid (g/L); pH (pH); PSC: probable stable color; PRI: phenolic ripeness index; TPI: total phenolic index; PCAF: probable color by end of alcoholic fermentation; ANT: anthocyanin content (CROMOENO data); and TAN: tannin content (CROMOENO data).

## 3. Results

### 3.1. Berry Characteristices

The results of the analysis of the sampled berries are shown in Table 3. Each row corresponds to a sampling zone (Z) represented in Figure 3b (by the green polygons) and is located within one of the repetitions transversal to the direction of the plantation rows (R).

### 3.2. Correlation Analysis

Table 4 shows the Pearson linear correlations for the chemical properties of must and vegetation indices derived from the RGB images.

Statistically significant correlations were found for five quality variables (BW, MA, AAN, PRI, and TPI) with nine vegetation indices (GR, $BGI_2$, GLI, VARI, VEG, GRVI, MGRVI, RGBVI2, and RGBVI3), as follows:

BW: Inverse correlation with the GR ($-0.76$) and direct correlations with the RGBVI2 (0.77), VARI (0.76), GRVI (0.76), MGRVI (0.76), GLI (0.72), RGBVI3 (0.72), and VEG (0.71) ($p < 0.01$ for all cases).

MA: Inverse correlations with the MGRVI ($-0.66$) ($p < 0.01$) and VARI ($-0.67$) ($p < 0.05$), RGBVI3 ($-0.67$) ($p < 0.05$), GRVI ($-0.66$) ($p < 0.05$), RGBVI2 ($-0.65$) ($p < 0.05$), and $BGI_2$ ($-0.59$) ($p < 0.05$), and direct correlation with the GR (0.66) ($p < 0.05$).

AAN: Inverse correlations with the VARI ($-0.59$), GRVI ($-0.59$), MGRVI ($-0.59$), and RGBVI2 ($-0.58$) ($p < 0.05$), and direct correlation with the GR (0.58) ($p < 0.05$ for all cases).

PRI: Inverse correlation with the GR ($-0.68$) and direct correlations with the VARI (0.69), GRVI (0.69), MGRVI (0.69), RGBVI2 (0.68), RGBVI3 (0.66), GLI (0.65), and VEG (0.64) ($p < 0.05$ for all cases).

TPI: Direct correlations with the RGBVI3 (0.62), VARI (0.59), GRVI (0.59), and MGRVI (0.59) ($p < 0.05$ for all cases).

### 3.3. Regresion Analysis

Table 5 presents the linear regression equations, coefficients of determination ($R^2$), and root mean square errors (RMSE) for statistically significant correlations between any five quality variables and any of the nine vegetation indices cited above. The best linear regressions are described below (RMSE is expressed in the units corresponding to each variable, i.e., kg$\cdot 10^{-3}$, g/L or mg/L):

BW. The best result was obtained for the RGBVI2, with $R^2 = 0.59$ and RMSE = 4.3, followed by the VARI, with $R^2 = 0.58$ and RMSE = 4.3, then by the GRVI ($R^2 = 0.58$ and RMSE = 4.3), MGRVI ($R^2 = 0.57$ and RMSE = 4.3), and GR ($R^2 = 0.51$ and RMSE = 4.4). The poorest results were for the GLI ($R^2 = 0.51$ and RMSE = 4.7), VEG ($R^2 = 0.50$ and RMSE = 4.7), and RGBVI3 ($R^2 = 0.47$ and RMSE = 4.9).

MA. The best results were obtained using the VARI ($R^2 = 0.44$ and RMSE = 0.13), GRVI, and MGRVI ($R^2 = 0.44$ and RMSE = 0.13 in both cases), followed by the GR and RGBVI3 ($R^2 = 0.43$ and RMSE = 0.13 in both cases), then by the RGBVI2 ($R^2 = 0.42$ and RMSE = 0.14). The poorest result, although statistically significant, was obtained for the $BGI_2$ index ($R^2 = 0.35$ and RMSE = 0.14).

AAN. Broadly similar results were obtained for several indices. $R^2 = 0.35$ for the VARI, GRVI, and MGRVI, RMSE = 37.5 for the VARI, and RMSE = 37.6 for both the GRVI and MGRVI. Results for the GR and RGBVI2 were $R^2 = 0.34$ and RMSE = 37.8 and $R^2 = 0.34$ and RMSE = 37.8, respectively.

PRI. The best results were obtained for the VARI, GRVI, and MGRVI (each $R^2 = 0.48$, and RMSE = 0.04 for the VARI and RMSE = 0.04 each for the GRVI and MGRVI), followed by the RGBVI2 and GR (each $R^2 = 0.47$, and RMSE = 0.04 for the RGBVI2 and RMSE = 0.0434 for the GR), then by the GLI ($R^2 = 0.42$ and RMSE = 0.05) and VEG ($R^2 = 0.41$ and RMSE = 0.05). The poorest result was obtained for the RGBVI3 ($R^2 = 0.39$ and RMSE = 0.05).

TPI. The best result was obtained for the RGBVI3 ($R^2 = 0.38$ and RMSE = 1.01), followed by the VARI ($R^2 = 0.35$ and RMSE = 1.03), then by the GRVI ($R^2 = 0.34$ and RMSE = 1.04)

and MGRVI ($R^2$ = 0.34 and RMSE = 1.04). The poorest result was obtained for the RGBVI2 ($R^2$ = 0.32 and RMSE = 1.06).

**Table 5.** Linear regression equations, determination coefficients ($R^2$), and root mean square errors (RMSE), for statistically significant correlations between five must quality variables and nine vegetation indices.

| | GR | GRVI | MGRVI | VARI | BGI$_2$ | VEG | GLI | RGBVI2 | RGBVI3 |
|---|---|---|---|---|---|---|---|---|---|
| **BW** | | | | | | | | | |
| Equation | $1x - 1.1 \times 10^{-2}$ | $1x - 3.1 \times 10^{-3}$ | $1x - 5.5 \times 10^{-3}$ | $1x - 6.1 \times 10^{-3}$ | | $1x - 1 \times 10^{-3}$ | $1x - 6 \times 10^{-4}$ | $1x - 9 \times 10^{-4}$ | $1x - 5 \times 10^{-4}$ |
| $R^2$ | 0.57 | 0.58 | 0.58 | 0.58 | | 0.50 | 0.51 | 0.59 | 0.47 |
| RMSE (kg·$10^{-3}$) | 4.4 | 4.3 | 4.3 | 4.3 | | 4.7 | 4.7 | 4.3 | 4.9 |
| **MA** | | | | | | | | | |
| Equation | $1x - 4 \times 10^{-6}$ | $1x - 3 \times 10^{-5}$ | $1x - 6 \times 10^{-6}$ | $1x - 2 \times 10^{-5}$ | $1x - 4 \times 10^{-5}$ | | | $1x - 6 \times 10^{-5}$ | $1x - 4 \times 10^{-5}$ |
| $R^2$ | 0.43 | 0.44 | 0.44 | 0.44 | 0.35 | | | 0.42 | 0.43 |
| RMSE (g/L) | 0.13 | 0.14 | 0.13 | 0.13 | 0.14 | | | 0.14 | 0.13 |
| **AAN** | | | | | | | | | |
| Equation | $1x - 9.3 \times 10^{-3}$ | $1x - 9.3 \times 10^{-3}$ | $1x - 6.2 \times 10^{-3}$ | $1x - 1.8 \times 10^{-2}$ | | | | $1x - 4 \times 10^{-4}$ | |
| $R^2$ | 0.34 | 0.35 | 0.35 | 0.35 | | | | 0.34 | |
| RMSE (mg/L) | 37.8 | 37.6 | 37.6 | 37.5 | | | | 37.8 | |
| **PRI** | | | | | | | | | |
| Equation | $1x + 4 \times 10^{-5}$ | $1x + 5 \times 10^{-5}$ | $1x - 5 \times 10^{-4}$ | $1x - 2 \times 10^{-6}$ | | $1x - 5 \times 10^{-5}$ | $1x - 3 \times 10^{-6}$ | $1x + 5 \times 10^{-6}$ | $1x - 7 \times 10^{-5}$ |
| $R^2$ | 0.47 | 0.48 | 0.48 | 0.48 | | 0.41 | 0.42 | 0.47 | 0.39 |
| RMSE | 0.04 | 0.04 | 0.04 | 0.04 | | 0.05 | 0.05 | 0.04 | 0.05 |
| **TPI** | | | | | | | | | |
| Equation | | $1x - 4 \times 10^{-4}$ | $1x - 6 \times 10^{-2}$ | $1x - 3 \times 10^{-4}$ | | | | $1x + 5 \times 10^{-4}$ | $1x + 4 \times 10^{-4}$ |
| $R^2$ | | 0.34 | 0.34 | 0.35 | | | | 0.32 | 0.38 |
| RMSE | | 1.04 | 1.04 | 1.04 | | | | 1.06 | 1.01 |

$R^2$: coefficient of determination; RMSE: root mean square error. **Vegetation indices.** GR: simple red-green ratio; GRVI: green-red vegetation index; RGBVI: RGB-based vegetation index; MGRVI: modified green-red vegetation index; VARI: visible atmospherically resistant index; BGI$_2$: simple blue-green ratio; VEG: vegetativen; GLI: green leaf index; ExG: excess green index; NGBDI: normalized green-blue difference index; RGBVI2: RGB-based vegetation index 2; RGBVI3: RGB-based vegetation index 3. **Must quality variables.** BW: 100-berry weight (kg $10^{-3}$); MA: malic acid (g/L); AAN: alpha amino nitrogen (mg/L); PRI: phenolic ripeness index; and TPI: total phenolic index.

Figure 5 depicts scatter plots of predicted versus reference values for the best correlations (significance $p > 0.05$) between quality variables and vegetation indices: BW with the RGBVI2 ($R^2$ = 0.59), MA with the VARI ($R^2$ = 0.44), ANN with the VARI ($R^2$ = 0.35), PRI with the VARI ($R^2$ = 0.48), and TPI with the RGBVI3 ($R^2$ = 0.38).

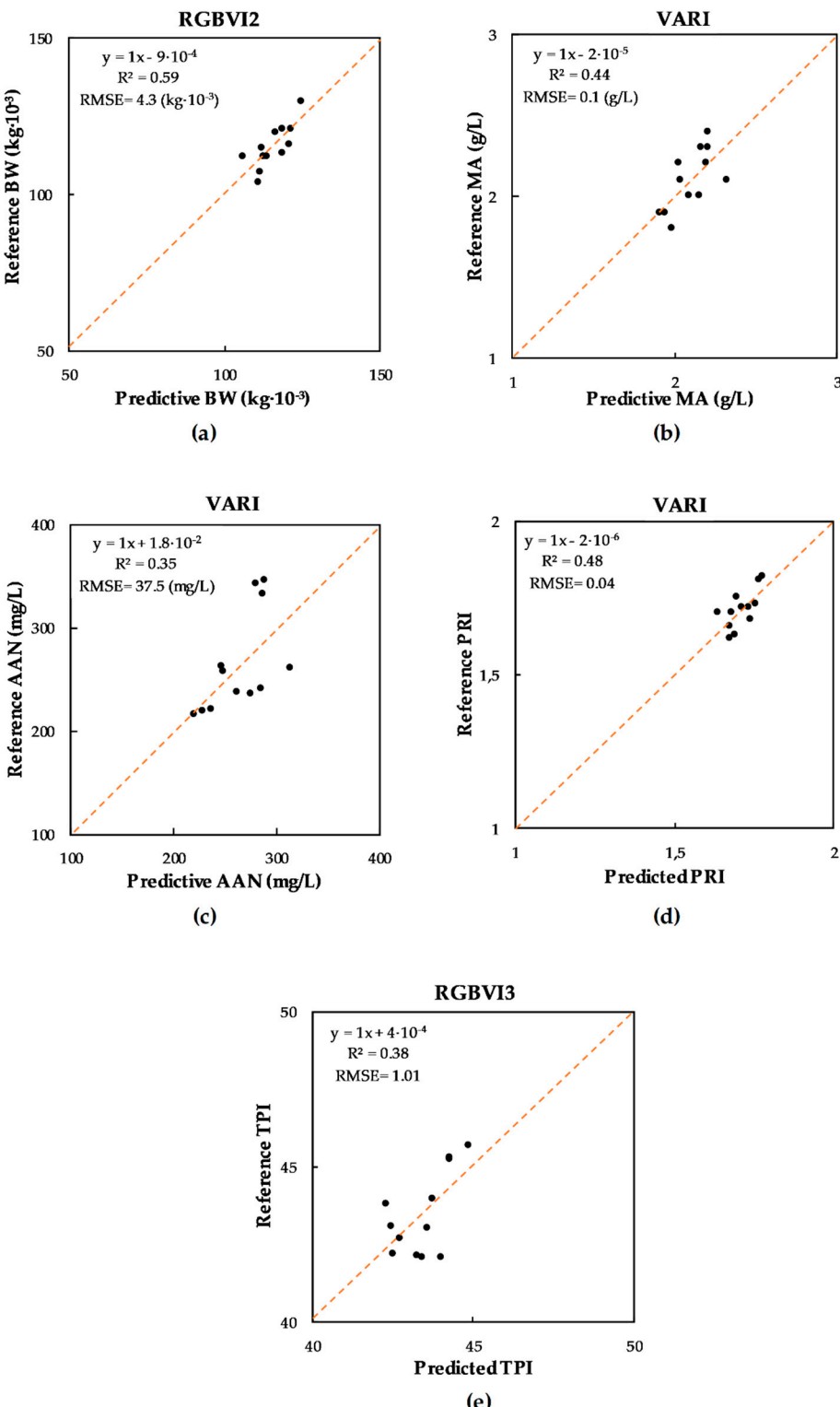

**Figure 5.** Linear relationship between must properties as measured by laboratory analysis and as predicted by the vegetation indices. (**a**) BW and RGBVI2. (**b**) MA and VARI. (**c**) AAN and VARI. (**d**) PRI and VARI. (**e**) TPI and RGBVI3. BW: 100-berry weight (kg·$10^{-3}$); MA: malic acid (g/L); AAN: alpha amino nitrogen (mg/L); PRI: phenolic ripeness index; and TPI: total phenolic index. The 1:1 line is indicated in each figure.

### 3.4. Spatial Variability Maps

Spatial variability maps of must quality variables were calculated (Figure 6) based on the best linear regression equations, as shown in Table 5. Correspondences between variables and vegetation indices were as follows—BW with RGBVI2, MA with VARI, AAN with VARI, PRI with VARI, and TPI with RGBVI3, labelled in Figure 6 from (a) to (e), respectively.

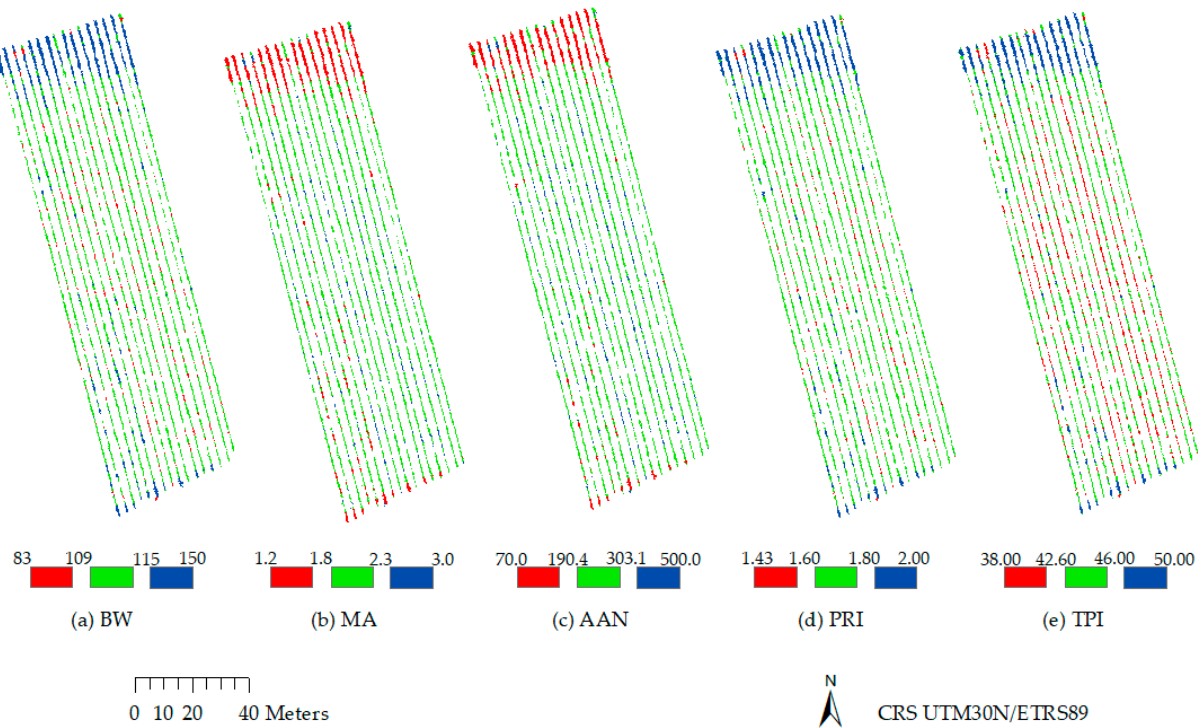

**Figure 6.** Spatial variability maps of must quality variables. (**a**) BW with RGBVI2, (**b**) MA with VARI, (**c**) AAN with VARI, (**d**) PRI with VARI, and (**e**) TPI with RGBVI3. The maps were classified in red, green and blue categories (low, intermediate and high values, respectively) using the natural breaks classification method. BW: 100-berry weight (kg·$10^{-3}$); MA: malic acid (g/L); AAN: alpha amino nitrogen (mg/L); PRI: phenolic ripeness index; and TPI: total phenolic index.

Referring first to maps (a), (b), (c), and (d) in Figure 6, one zone clearly corresponded to the north of the vineyard, where values (in the units corresponding to each variable, i.e., kg·$10^{-3}$, g/L, or mg/L) were highest for BW (between 115 and 150) and PRI (between 1.80 and 2.00), and lowest for MA (between 1.2 and 1.8) and AAN (between 70.0 and 190.4). The central—and largest—part of the vineyard corresponded to intermediate values for BW (109 to 115), MA (1.8 to 2.3), AAN (190.4 to 303.1), and PRI (1.60 to 1.80). Some small areas had other categories interspersed, with the lowest values in the BW (a) and PRI (d) maps, and with the highest values in the MA (b) and AAN (c) maps.

Map (e) in Figure 6, corresponding to TPI spatial distribution according to the RGBVI3, shows three well differentiated classes—the highest values (46.00–50.00) towards the north of the vineyard, intermediate value (42.60–46.00) in the central area, and the lowest values (38.00–42.60), also in the central area, but interspersed with values corresponding to the intermediate class.

Zoning for effective management purposes for maps (a), (b), (c), and (d) in Figure 6 would suggest two zones—to the north of the vineyard, with higher values for BW and PRI, and lower values for MA and AAN, and the rest of the vineyard, with intermediate values for those variables. As for TPI (Figure 6e), possible zones were in the north and the extreme south of the vineyard, which showed the highest TPI values, and a central area, which showed the lowest and intermediate TPI values.

## 4. Discussion

The main objective of this study was to evaluate RGB images acquired from low-cost UAV and RGB vegetation indices extracted from those images for the purpose of predicting must quality variables. Most classic vegetation indices were developed for images captured from satellite platforms. These indices used green, red, and infrared bands, but not blue, due to the distortion produced by the atmosphere in short-wave radiation. However, in UAV imagery, atmospheric interference was much lower, so the blue band could be included to provide information related to biophysical variables.

It was possible to obtain an orthomosaic map with centimetric resolution derived from the images. The sensor used for taking images was previously characterized [38], which helped radiometrically calibrate the orthomosaic map, using the empirical linear method [37]. Correlations with the point values for the quality variables (i.e., BW, MA, TaA, AAN, EAN, GA, TSS, ToA, pH, PSC, PRI, TPI, PCAF, ANT, and TAN) were analyzed in the laboratory with RGB vegetation indices.

The results in Table 4 show that BW, MA, AAN, PRI, and TPI obtained the best results in terms of correlations with the vegetation indices. It was found that BW, a quantitative biophysical parameter related to berry yield and vigor, showed very good correlations with eight of the 12 vegetation indices, very positively correlated (0.77) with the RGBVI2 developed for this study, and negatively correlated ($-0.76$) with the GR index. Results for BW research carried out by other authors [24,26] were not as promising as those obtained here. Bonilla et al. [24] used infrared (IR)-based indices and calibrated aerial images (GSD = 0.5 m/pixel) previously segmented by an NDVI-based threshold to separate the canopy from vegetation [24], while Matese et al. [26] used calibrated UAV images with centimetric resolution similar to ours (GSD = 0.05 m/pixel) to calculate a volume index (VI) for vegetation; for the former study, Pearson correlation between the average reflectances around each sampling point (in a 1.2 m buffer) was a positive 0.51 [24], while for the latter study, the relationship between the Moran's index and the VI was a negative, although non-significant, $-0.39$ [26]. Similar but slightly better results were reported by Di Gennaro et al. [32]; Pearson correlation values were 0.82 and 0.89, respectively, for the IR images of the Sentinel 2 platform (GSD = 10 m/pixel) and the UAV-captured IR images (GSD = 0.03 m/pixel), previously calibrated (linear method) [38] and segmented by an unsupervised algorithm and by the Otsu threshold. Note that in investigations where variables were analyzed using satellite images and the NDVI, it was shown—despite the very good correlations—that the results for variables related to quantitative biophysical parameters, such as BW as analyzed in this work, depend on the dataset and the acquisition time [19], and that the applicability of RGB indices is limited to certain growth stages [43,50].

Regarding the grape composition traits, MA obtained good correlations with seven RGB vegetation indices, but primarily VARI ($-0.67$), GRVI with the same sign ($-0.66$), and the GR ratio, with a positive sign (0.66). These studies found that in the analyzed correlation of this variable with images, the results were no better. For the MA correlations with aerial images taken from conventional airplanes, IR images with a resolution of 0.2 m were calibrated and processed to reduce the radiometric effect of the inter-row areas using a filter [22]. It was detected that the highest concentrations were located in areas of low vigor, for a correlation value of $-0.34$, and in years when grape ripening was not affected by water restrictions [22]. In contrast, in our research using conventional UAV-captured RGB images, correlations were significantly better, suggesting that the VARI, GRVI, and GR indices might be related to MA, water potential, and vigor. Our results also surpassed those of Di Gennaro et al. [27], who, using thermal data and calibrated IR UAV-captured high-resolution images (GSD = 0.04 m/pixel), reported inverse (although not significant) correlations for the areas of low vigor.

The results of the analysis of the TPI presented good correlations with five RGB indices, showing the highest positive correlation (0.62) with the RGBVI3 proposed for this study and a negative correlation of $-0.57$ for the GR ratio. Previous research into TPI correlation

with vegetation indices based on the IR spectrum reported negative correlations with vigor that were related to high polyphenol content [6,22–24], and in some cases, reported TPI as dependent on water content [22]. The methods used in the cited works were based on digital image processing to calibrate and reduce the effect of reflectance in vineyard areas of no interest, such as shaded areas or inter-row soil [22–24]—as was done in our research. The highest correlation with this variable was reported by Fiorillo et al. [23] (−0.85), who used a convolution filter to eliminate inter-row areas in resampled images with a geometric resolution of 0.6 m/pix (3 × 3). Other high correlations were reported, for instance, by Ferrer et al. [22] (−0.6), who reported inverse correlations with the NDVI index (−0.6) in aerial images (GSD = 0.2 m/pix) in which the shadows and inter-row areas were masked, by Lamb et al. [6] (−0.59), where an IR scenario with a resolution 0.6 m/pix was resampled at a pixel size of 5 × 5; and, finally, by Bonilla et al. [24] (−0.41), as described in the previous paragraph.

Phenolic maturation indicated the optimal state of ANT and TAN compounds for harvesting. In the case of this variable, the greatest positive (0.69) and negative (−0.68) correlations were with the VARI and the GR ratio, respectively. For AAN, the greatest negative (−0.59) and positive (0.58) correlations were with the VARI and the GR ratio.

As for the analyzed variables that did not present significant correlations, note that, in the literature, pH, TSS, ToA, TAN, and ANT were related to the NDVI, using calibrated multispectral images [23–25]. TSS as analyzed in our study correlated with the ExG index (0.53), exceeding the −0.43 reported elsewhere [24] for the NDVI. Although a strong correlation with TSS was reported for the NDVI (−0.77) [23], obtaining information using a conventional RGB camera saved the cost of using a calibrated IR camera. The results of our study showed that GRVI, MGRVI, and RGBVI2 were inversely related to pH, with a value (−0.52) that is an improvement on values previously reported for the NDVI—a direct correlation of 0.43 [24] and an inverse correlation of −0.48 [25]. The −0.52 obtained for GRVI, MGRVI, and RGBVI2 was close to the 0.55 reported for the pH correlation with the NDVI [23]. Regarding the TAN and ANT variables, the correlations in our study were found to be quite weak. For TAN, the best correlation was with VEG (0.15). In other studies, strong correlations were obtained for TAN (−0.85) [23] for the IR band. For ANT, the best correlation was inverse to ExG (−0.37), with values close to the −0.75 and −0.65 published elsewhere [23,25].

The indices proposed in this work include the blue band. Chlorophyll a and b absorption peaks were in the bands of 420 nm, 490 nm and 660 nm, and 435 nm and 643 nm, respectively [51]. Therefore, both indices were related to the color and chlorophyll content of the leaves.

Remote sensing estimation of chlorophyll content in vines was previously studied in viticulture [52,53]. Kandylakis et al. [20] related total polyphenol content with chlorophyll content using the index proposed in Gitelson et al. 2006 [54]. The new index proposed in this work, RGBVI3, which correlated with the TPI, could be used as an indicator of chlorophyll b content since it included the blue band.

With the RGBVI2 and RGBVI3 indices, it was possible to identify vineyard areas with larger grapes and higher polyphenol content (Figure 6), corroborating Ferrer et al. [22], who also reported high correlation between those variables.

## 5. Conclusions

Drones (UAVs) are now widely used for remotesensing operations in viticulture, as they enable the acquisition of high-resolution spatial images that can discriminate between the vegetation canopy, bare soil, and shaded zones, yielding pure vine leaf pixels. As a means to easily and inexpensively collect and estimate must quality data from in situ measurements, we used a drone with a conventional camera to capture images with which to calculate visible band vegetation indices.

The correlation results indicated that the must variables that are important for producing quality wines could be related to the RGB bands in conventional images, which are

used to create vegetation indices and predict quality variables related to wine production. Statistical analysis showed that RGB indices derived from aerial images captured by UAVs could be good indicators of quality variables related to wine production. Thus, BW could be predicted with our newly developed RGBVI2; MA, AAN, and PRI with the VARI; and TPI could be predicted with our newly developed RGBVI3. There is also the possibility of using the Pearson correlation to estimate other key variables in quality wine production, namely, pH and TSS, as a relationship with spectral bands in the region was also found, although it was less significant than for the previously mentioned variables.

The fact that the variables that presented outstanding correlations were related to grape ripening and vigor suggests the potential of using conventional images to monitor harvests and to zone different sections of the vineyards for specific treatment or management. The use of RGB images captured by drones with integrated sensors is an inexpensive and accessible way to obtain spectral information from crops. Note, however, that radiometric sensor characterization using the empirical linear method requires a high-cost sensor. Using digital image processing techniques, digital levels of the visible spectrum could be transformed into reflectance to create vegetation indices aimed at obtaining information on the spatial distribution of crop quality variables.

Future research will focus on corroborating our findings, using the proposed indices to estimate new variables (such as content in photosynthetic pigments) and to validate the results in other grape varieties.

**Author Contributions:** Conceptualization: M.G.-F., E.S.-A. and J.R.R.-P.; methodology: M.G.-F., E.S.-A. and J.R.R.-P.; software: E.S.-A.; formal analysis: M.G.-F. and E.S.-A.; writing—original draft preparation: M.G.-F., E.S.-A. and J.R.R.-P.; writing—review and editing: M.G.-F., E.S.-A. and J.R.R.-P.; visualization: E.S.-A. and J.R.R.-P.; supervision and project administration: J.R.R.-P. All authors have read and agreed to the published version of the manuscript.

**Funding:** This research was funded by the Education Department of the Junta de Castilla y León, grant number LE112G18.

**Institutional Review Board Statement:** Not applicable.

**Informed Consent Statement:** Not applicable.

**Data Availability Statement:** Data sharing not applicable.

**Acknowledgments:** This research was supported by funding from the Education Department of the Junta de Castilla y León-Spain. Grant number LE112G18, under a call for research by recognized research groups attached to public universities starting in 2018 (Order 20 November 2017). Marta García Fernández gratefully acknowledges financial support provided by the Fundación Carolina Rodríguez and Universidad de León.

**Conflicts of Interest:** The authors declare no conflict of interest. The authors declare that they have no known competing financial interests or personal relationships that could have appeared to influence the work reported in this paper.

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
