# Peer review of "High-Resolution Drone-Acquired RGB Imagery to Estimate Spatial Grape Quality Variability"

_agronomy, doi:10.3390/agronomy11040655_

Round 1

Reviewer 1 Report

accept

Author Response

Thanks for your review of this work. Your comments were very appropriate and have contributed greatly to my approach to this article and to the focus of future work.

Reviewer 2 Report

Dears Authors ,

Thank you for implementing the requests made previously. In this form the manuscript better explains some of the passages of your scientific research 

Author Response

(The authors gave the same response as above.)

Reviewer 3 Report

The paper reports the efforts of finding significant and useful correlations between must ripening parameters and drone-derived RGB vegetation indices. The implementation of such technique may represent a cost-effective and time-saving methodology for vineyard management. The authors proposed two novel vegetation indices, thus increasing the novelty of the study.

Keywords: I don’t think correlation and regression are crucial keywords for this manuscript, I’d suggest adding something related to remote sensing/drone/imagery.

The Introduction provides a good overview of the topic and previous application of UAVs imagery.

M&M are well and clearly described. However, I would expect an explanation of the criteria and reasons for the adoption of the new vegetation indices.

The results are clearly presented. I have some concerns about the maps provided in Figure 6. More details on these concerns are explained below.

The Discussion needs to be improved. Especially, I would expect to have some explanation and not only comparison with other studies. The authors should try to link their results with the indices. Why do they believe specific indices were related to must quality parameters? This would also enhance their effort in generating new vegetation indices: it would be an occasion to explain why they combined those specific bands in their custom vegetation indices. Moreover, I have some concerns about the maps showed in Figure 6. The maps show a very different spectral behavior of the north side of the vineyard. In my opinion, this may be due to the well-known heterogeneity of vineyard external borders. The authors should clarify this point.

The Conclusions could be improved by adding some information of future work to reduce the weaknesses of the study.

Overall, the quality of the manuscript is high, but I believe the Discussion should be improved before accepting it for publication.

Here are some specific comments:

Line 9: remove typo (“8”)

Line 46: there’s something missing, as there’s only one ending bracket

Line 52-53: this sentence is not clear; you should better explain what’s the link of must parameters with calcination and 42 °C

Figure 2: optional, it would be better to add the coordinates on the sides of the figure

Paragraph 2.4: could you add information on the overlap percentage?

Line 210: why don’t you call them GRBVI2 and GRBVI3?

Page 12, lines 4-16: there is a bit of confusion in the identification of the p-values. You should decide whether you want to display the highlight the p-values of each correlation coefficient (advisable) or just once.

Figure 5: the quality of this figure should be increased. Also Figure 6 could be improved

Author Response

Before responding to your comments and suggestions, I’d like to thank you for your review of this work, which has contributed greatly to my approach to this article and to the focus of future work.
